# Architecture of the Sema3A/PlexinA4/Neuropilin tripartite complex

Defen Lu[1,5], Guijun Shang[1,5], Xiaojing He[2,5], Xiao-chen Bai ⬤ [3,4✉] & Xuewu Zhang ⬤ [1,3✉]

Secreted class 3 semaphorins (Sema3s) form tripartite complexes with the plexin receptor and neuropilin coreceptor, which are both transmembrane proteins that together mediate semaphorin signal for neuronal axon guidance and other processes. Despite extensive investigations, the overall architecture of and the molecular interactions in the Sema3/plexin/neuropilin complex are incompletely understood. Here we present the cryo-EM structure of a near intact extracellular region complex of Sema3A, PlexinA4 and Neuropilin 1 (Nrp1) at 3.7 Å resolution. The structure shows a large symmetric 2:2:2 assembly in which each subunit makes multiple interactions with others. The two PlexinA4 molecules in the complex do not interact directly, but their membrane proximal regions are close to each other and poised to promote the formation of the intracellular active dimer for signaling. The structure reveals a previously unknown interface between the a2b1b2 module in Nrp1 and the Sema domain of Sema3A. This interaction places the a2b1b2 module at the top of the complex, far away from the plasma membrane where the transmembrane regions of Nrp1 and PlexinA4 embed. As a result, the region following the a2b1b2 module in Nrp1 must span a large distance to allow the connection to the transmembrane region, suggesting an essential role for the long non-conserved linkers and the MAM domain in neuropilin in the semaphorin/plexin/neuropilin complex.

[1] Department of Pharmacology, University of Texas Southwestern Medical Center, Dallas, TX, USA. [2] Key Laboratory of Molecular Biophysics of the Ministry of Education, College of Life Science and Technology, Huazhong University of Science and Technology, Wuhan, China. [3] Department of Biophysics, University of Texas Southwestern Medical Center, Dallas, TX, USA. [4] Department of Cell Biology, University of Texas Southwestern Medical Center, Dallas, TX, USA. [5] These authors contributed equally: Defen Lu, Guijun Shang, Xiaojing He. ✉email: xiaochen.bai@utsouthwestern.edu; Xuewu.zhang@utsouthwestern.edu

Semaphorins are the largest family of guidance molecules that transduce signal to control cell morphological changes, migration and directional growth. Specifically, semaphorin signaling plays essential roles in the development and functions of the nervous system and cardiovascular system by serving as guidance cues for developing neurons and blood vessels[1,2]. Loss-of-function of semaphorins leads to severe developmental defects in these systems and in some cases embryonic lethality[1,2]. Semaphorin-mediated signal also play important roles in regulating other processes, such as wound healing, immunity, and bone development[1–3].

Semaphorins are divided into eight classes based on sequence conservation. They are either cell surface attached or secreted proteins that exert their biological effects through binding and activating their cognate cell surface receptors. The semaphorin family members all contain a N-terminal Sema domain, characterized by a 7-bladed β-propeller fold, that mediates the interaction with the receptor[4–10]. In most semaphorins, the Sema domain is followed by a small cysteine-rich plexin-semaphorin-Integrin (PSI) domain and an immunoglobin-like (Ig-like) domain. Semaphorins typically form homodimers and in some cases heterodimers through interactions mediated by the Sema and Ig-like domains[4–10]. The region following the Ig-like domain is more diverse among the semaphorin family members. A conserved cysteine residue in the C-terminal tail region of class 3 semaphorins forms an interchain disulfide to covalently link the two subunits of the dimer[11,12].

The best characterized receptor of semaphorins are plexins, which are classified into four classes (A, B, C, and D)[1,2]. Plexins are type I single-pass transmembrane proteins that also contain a Sema domain at the N-terminus of the extracellular region, which binds the Sema domain in semaphorins. The Sema domain in plexins is followed by multiple PSI and IPT (Integrin, Plexin, and Transcription factor) domains, which connect to the transmembrane (TM) helix. The intracellular region of all plexins contains a membrane proximal juxtamembrane region (JM), a Rho GTPase-binding domain (RBD), and GTPase Activating Protein (GAP) domain[13–18]. In general, the binding of dimeric semaphorin leads to the formation of the active dimer of plexin, which activates the GAP activity of plexins specific for Rap GTPase that is essential for signaling[2,13,14,19,20]. The intracellular region of some plexins can interact with other proteins for signal transduction through additional pathways or intracellular trafficking[21–23].

An exception to the general mechanism described above is that secreted class 3 semaphorins binds class A plexins very weakly, and therefore cannot induce their activation independently[24–26]. In this case, the transmembrane protein neuropilin (Nrp1 or Nrp2) acts as an essential co-receptor to facilitate the semaphorin/plexin interaction. Neuropilin interacts with both semaphorin and plexin, leading to a stable 2:2:2 complex of semaphorin, plexin and neuropilin that is capable of triggering downstream signaling[7]. Nrp1 and Nrp2 are also type I transmembrane proteins, composed of a multi-domain extracellular region, a single transmembrane helix and a short cytoplasmic tail. The extracellular region contains from the N- to C-terminal region the a1, a2, b1, b2, and MAM (Meprin, A5, and Mu-phosphatase) domains. The a1 and a2 domains belong to the $Ca^{2+}$-binding CUB domain family, whereas the b1 and b2 are coagulation factor V/VIII homology domains[27–29]. The MAM domain exhibits a jellyroll topology and also binds $Ca^{2+}$.[30] The extracellular domains of neuropilin play major roles in stabilizing the semaphorin/plexin/neuropilin complex, while the intracellular region interacts with proteins that contribute to downstream signaling or regulate trafficking of neuropilin[2]. Another interesting aspect is that some class 3 semaphorins require both neuropilin and the simultaneous presence of two different class A

plexins for signaling[31,32]. Overexpression of one plexin in some cases could overcome this requirement[31,32]. These observations suggest that different plexin family members can heterodimerize and gain extra structural and functional features, the precise mechanisms of which are not well understood at present.

Extensive structural analyses of plexin have led to a general model on how dimeric semaphorin binds and induces the dimerization and activation of plexin[6,7,9,10,13,33,34]. While all the plexins use similar mechanisms of activation, there are some important differences for some plexin family members. For example, PlexinC1 has a distinct extracellular architecture, and therefore the overall shape of the semaphorin-induced dimer of PlexinC1 appears quite different from that of class A plexins[33]. A crystal structure of the a1 domain of Nrp1 in complex with PlexinA2 and Sema3A shows that the Nrp1-a1 domain stabilizes the complex by binding the Sema domains of both semaphorin and plexin simultaneously[7]. Interestingly, biochemical analyses presented in this study showed that the a1a2b1b2 domains together are required for the semaphorin/plexin/neuropilin complex formation, but the a2b1b2 domains are not resolved in the structure and therefore their roles in the complex remain unclear[7]. In addition, the C-terminal tail region of class 3 semaphorins contributes to the complex formation by interacting with neuropilin[25,26,35,36]. Class 3 semaphorins could be cleaved at multiple sites by furin or furin-like proteases, which recognize a degenerate K/R-XX-R (K, lysine; R, arginine; X, any residue) motif[37]. Of particular interest, furin-cleavage of the C-terminal poly-basic region of class 3 semaphorins results in a C-terminal arginine residue that serves as a high-affinity ligand for a pocket on the b2 domain of neuropilin[37–41]. Notably this pocket on the b2 domain of neuropilin has recently been suggested to bind a C-terminal arginine-containing motif as a result of furin-mediated cleavage of the spike protein of SARS-CoV-2, and thereby contributing to viral infection[42,43].

Taken together, neuropilin interacts with semaphorin and plexin through multiple binding sites. How these multiple sites work together in the context of the 2:2:2 semaphorin/plexin/neuropilin assembly is not known due to lack of structure analyses of an intact complex. We reconstituted the complex using the a1a2b1b2 domains of Nrp1, the near full-length extracellular regions of PlexinA4 and Sema3A. We determined the structure of this complex to 3.7 Å resolution by using cryo-electron microscopy (Cryo-EM) single particle reconstruction. The structure reveals the overall architecture of the 2:2:2 complex and the details of the multiple interfaces among the subunits that stabilize this large assembly. In addition, this architecture reveals previously unappreciated important roles for the long and non-conserved interdomain linker regions in both neuropilin and semaphorin in the complex formation.

## Results

**Structure determination of the Sema3A/PlexinA4/Nrp1 complex.** The results from early cell-based assays have demonstrated that class 3 semaphorins at low- or subnanomolar concentrations can bind their plexin/neuropilin receptors and induce cell collapse that reflect their repulsive guidance activity, indicating of high-affinity interactions[24–26,44]. However, thorough in vitro binding assays with surface plasma resonance have shown that the dissociation constants between individual pairs of the Sema-PSI domains of Sema3A, the Sema-PSI1-IPT1-PSI2 domains of PlexinA2 and the a1a2b1b2 domains of Nrp1 range from double-digit micromolar to undetectable[7]. The binding affinity of PlexinA2 to Sema3A and Nrp1 both immobilized on the same sensor chip is stronger ($K_d$ value of 6 μM), but still three orders of magnitude weaker than those indicated by the cell-based assays.

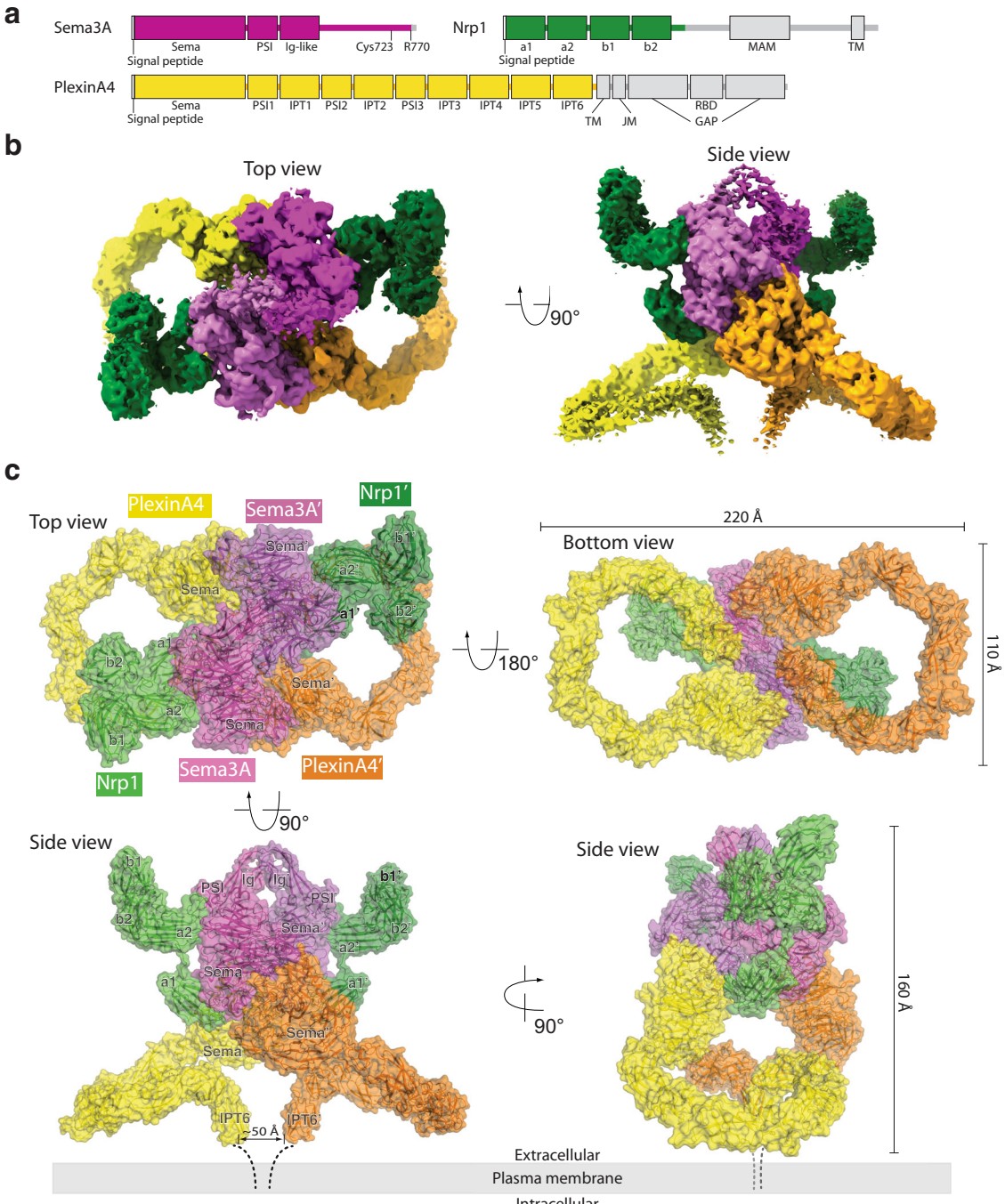

**Fig. 1 Overall structure of the Sema3A/PlexinA4/Nrp1 complex. a** Domain structures of mouse Sema3A, PlexinA4, and Nrp1. Colored parts indicate boundaries of constructs used in the structural analyses. **b** Overview of the cryo-EM map of the Sema3A/PlexinA4/Nrp1 complex. The color scheme is the same as **a**. **c** Atomic model of the Sema3A/PlexinA4/Nrp1 complex based the cryo-EM reconstruction.

This discrepancy is in part owing to the fact that the in vitro binding assays used a Sema3A construct lacking the C-terminal poly-basic tail, which is processed by furin-like proteases to generate a motif with a C-terminal arginine residue that binds the b1 domain of Nrp1 with high affinity (reported $K_d$ values in the low nanomolar to micromolar range)[25,38,39]. In our effort of reconstitution of the semaphorin/plexin/neuropilin complexes, we therefore used constructs of semaphorins containing the C-terminal tail but truncated at one of the furin-cleavage sites, equivalent to the furin-processed mature forms (Fig. 1a). We also introduced to the semaphorin constructs the mutation corresponding to A106K in Sema3A, which was rationally designed to

enhance the binding to plexin[45]. For neuropilin, we tested the full-length extracellular region and constructs with the MAM domain truncated, as previous studies have shown that the MAM domain is not directly involved binding either semaphorin or plexin[7,36,44,46].

Among the many combinations of semaphorin, plexin and neuropilin proteins tested, the complex formed by co-expression of mouse Sema3A ending at Arg770 with the A106K mutation (Sema3A (21–770)-A106K), the MAM-domain truncated version of Nrp1 (Nrp1-a1a2b1b2) and the full-length extracellular region of PlexinA4 was stable in gel filtration chromatography and showed relatively homogenous particles in our cryo-EM

**Table 1 Data collection and model refinement statistics.**

| Data collection and processing | |
|---|---|
| Magnification | 46,730 |
| Voltage (kV) | 300 |
| Electron exposure (e⁻/Å²) | 50 |
| Defocus range (μm) | 1.5–2.5 |
| Pixel size (Å) | 1.08 |
| Symmetry imposed | C2 |
| Map resolution (Å) | 3.7 |
| FSC threshold | 0.143 |
| Model refinement | |
| Initial model used (PDB code) | 5l5k, 4gz8 & 2qqk |
| Model resolution (Å) | 4.2 |
| FSC threshold | 0.5 |
| Map sharpening B factor (Å²) | −30 |
| Model composition | |
| Non-hydrogen atoms | 30,898 |
| Protein residues | 4228 |
| Ligands | 4 |
| B factors (Å²) | |
| Protein | 260 |
| Ligand | 216 |
| R.m.s. deviations | |
| Bond length (Å) | 0.004 |
| Bond angle (°) | 0.7 |
| Validation | |
| Molprobity score | 2.06 |
| Clashscore | 17.4 |
| Poor rotamers (%) | 1 |
| Ramachandran plot | |
| Favored (%) | 95.6 |
| Allowed (%) | 4.4 |
| Outliers (%) | 0 |

screen (Supplementary Figs. 1, 2). We collected a cryo-EM dataset for this complex on a Titan Krios 300 kV microscope and obtained a 3D reconstruction at overall resolution of 3.7 Å (Supplementary Fig. 2 and Table. 1). The cryo-EM map reaches high local resolutions for the Sema domains of both Sema3A and PlexinA4, whereas the C-terminal domains of both Sema3A and PlexinA4 are less well resolved (Supplementary Fig. 3). All the four domains (a1, a2, b1, and b2) in the Nrp1 construct are present in the reconstruction, although the density for the b1 and b2 domains is weaker than that of a1 and a2 domains. Model building was initiated by docking available crystal structures of various parts of the complex, followed by manual building and real-space refinement. The final model contains all the structured domains of the three proteins, but with various segments in some domains and many sidechains not included due to lack of clear density (See method for details).

**Overall architecture of the Sema3A/PlexinA4/Nrp1 complex and the activation mechanism of plexin.** The cryo-EM structure shows a 2-fold symmetric 2:2:2 complex of the extracellular regions of Sema3A, PlexinA4, and Nrp1, with an overall dimension of ~220 × 160 × 110 Å (Fig. 1b, c and Supplementary movie. 1). From a sideview, the overall shape of the complex resembles a frog. The Sema domains of Sema3A and PlexinA4 and the a1 domain of Nrp1 together form the central compact part of the complex that can be considered the body of the frog. The Ig-like domains from the two Sema3A subunits form the head, while the two a2b1b2 modules of Nrp1 resemble two front limbs. The extracellular region of PlexinA4 adopts the highly curled ring-like shape as seen before[34,47], resembling two rear limbs that extend sideways but fold back and converge at the

membrane proximal IPT6 domain, which represents the foot tucked near the rear end of the body.

The binding mode between the Sema3A dimer and the two PlexinA4 molecules in the complex is similar to those in all the previous structures of semaphorin/plexin complexes, with or without neuropilin (Supplementary Fig. 5)[6,7,9,10,33]. However, the cryo-EM structure provides a picture of the entire extracellular region of the 10-domain class A plexin in a ligand-induced active dimeric complex. A superimposition based on the N-terminal Sema domain of the PlexinA4 extracellular region in the complex with the previously reported crystal structure of PlexinA4 in the apo-state shows that their overall ring-shapes are very similar (Supplementary Fig. 6a)[47]. There are only small differences in the relative orientations among the domains following the Sema domain, which result in a slightly more tightly curled shape of the apo-PlexinA4 crystal structure compared with that in the cryo-EM structure. Such similar conformations of PlexinA4 obtained from completely different conditions and states highlight the remarkable rigidity of the extracellular region of PlexinA4 despite containing a total of 10 domains that are organized in a consecutive fashion and lack long-range interactions among non-neighboring domains. The rigid ring-shape of PlexinA4 in combination with the Sema3A/PlexinA4 binding mode leads to close juxtaposition and roughly parallel relative orientation between the two membrane proximal IPT6 domains from the two PlexinA4 molecules in the complex (Fig. 1b, c), consistent with the previous models based on the structures of the plexin extracellular region docked to semaphorin dimers[47]. The membrane proximal IPT6 domain is connected to the transmembrane region of plexin through a short linker (~10 residues). The ~20 residues from the two copies of the linker can span the distance (~50 Å) between the C-termini of the two IPT domains in the dimeric complex. The architecture of the Sema3A/PlexinA4/Nrp1 extracellular region complex is therefore poised to induce the formation of the intracellular active dimer of plexin for signaling, in a similar manner as that seen in PlexinC1 activated by the viral semaphorin-mimic A39R[13,33].

**Role of Nrp1 in the Sema3A/PlexinA4/Nrp1 complex.** Each subunit of the Sema3A dimer binds one PlexinA4 molecule to form a 2:2 complex where the two PlexinA4 molecules do not contact each other (Fig. 1). The 2:2 Sema3A/PlexinA4 complex can also be viewed as two Sema3A/PlexinA4 heterodimers, which are held together by interactions between the two Sema3A molecules as well as the interactions contributed by Nrp1. The a1 domain of Nrp1 wedges between the Sema3A molecule from one of the heterodimers and the PlexinA4 molecule from the second heterodimer, helping glue the two heterodimers together. This binding mode of the Nrp1-a1 domain with semaphorin and plexin is similar to that observed in the previous crystal structure of the complex of the Sema3A-Sema-PSI domains, the PlexinA2-Sema-PSI1-IPT1-PSI2 domains and the Nrp1-a1 domain[7]. The higher resolution of the cryo-EM structure presented here (3.7 Å resolution compared with the 7.0 Å resolution of the crystal structure) reveals more details of the binding interfaces (see more below). The a2b1b2 domains of Nrp1, adopting the characteristic triangular shape[7,28], is placed on the side of Sema3A above both the a1 domain and PlexinA4. The a2 domain of Nrp1 makes an interface with the Sema domain of Sema3A, which has not been observed before (See more below). Presumably, the b1 domain binds the C-terminal arginine motif of Sema3A, but this interaction is not resolved in the cryo-EM map and therefore not included in the atomic model.

The position of the a2b1b2 module of Nrp1 in the complex indicates that these domains are high above the surface of the

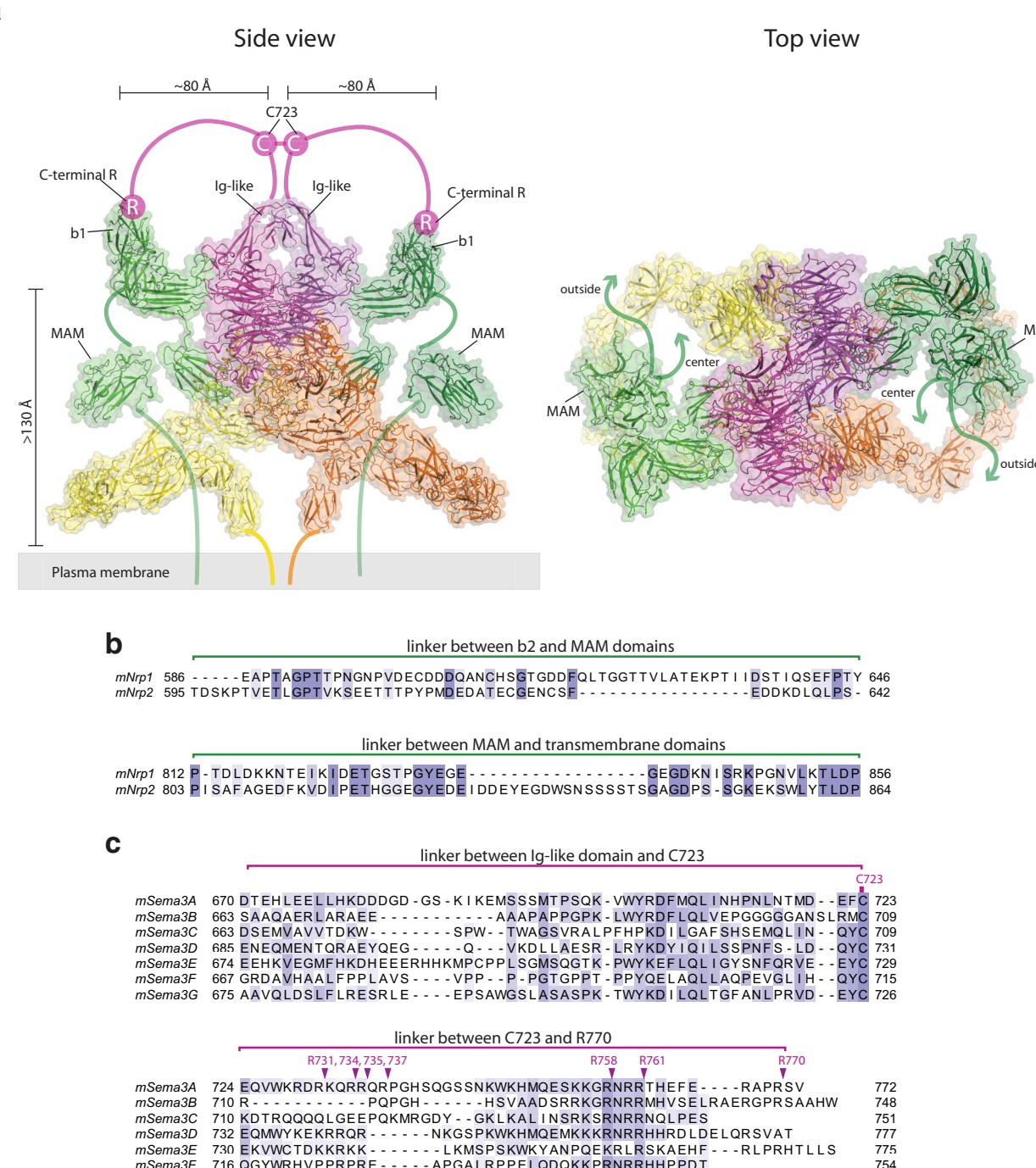

**Fig. 2 Importance of the linker regions in the formation of the Sema3A/PlexinA4/Nrp1 complex. a** Roles of the linkers in connecting various parts of the proteins in the tripartite complex on the cell surface. The crystal structure of the MAM domain of Nrp1 (PDB ID: 5I73) is docked between the a2b1b2 module and the plasma membrane. Thick lines represent the linker regions. The arrows in the right panel (top view) indicate that the MAM-linker region may use two alternative paths (through the center or outside of the PlexinA4 ring) to connect to the TM region of Nrp1. **b** Sequence alignment of the b2-MAM and MAM-TM linkers from mouse Nrp1 and Nrp2. **c** Sequence alignment of the tail regions of mouse class 3 semaphorins. Putative furin-cleavage sites are highlighted.

plasma membrane where both PlexinA4 and Nrp1 are anchored through their respective transmembrane regions (Fig. 1). As shown in Fig. 2, the region following the b2 domain of neuropilin, which includes the b2-MAM linker, the MAM domain and the MAM-TM linker, could reach the transmembrane region by passing through the center of the PlexinA4 ring, or either side of the ring. In both scenarios, this region needs to span a large distance (over 130 Å). The MAM domain adopts a compact structure, with the distance between its N- and C-termini of ~30 Å[30]. Therefore, the b2-MAM and MAM-TM linkers must cover ~100 Å distance to allow the formation of the semaphorin/plexin/neuropilin complex on the cell surface (Fig. 2a). Consistent with this requirement, the two linkers in both mouse Nrp1 and Nrp2 are rather long, containing ~100 residues in total (Fig. 2b).

A search in the Uniport database showed that the linkers in neuropilin from other species are in general of similar length. The linkers are predicted to be extended with no secondary structures, and much less conserved compared to other regions of neuropilin. Interestingly, the b2-MAM linker in Nrp2 is ~13-residue shorter than that in Nrp1, whereas the MAM-TM linker in Nrp2 is ~16-residue longer (Fig. 2b). These observations suggest that the total length of the two linkers together is important, while the distribution of residues in the two linkers may be not, consistent with the idea that these linkers serve as spacers, rather than mediate specific protein–protein interactions. The differential lengths of the two linkers likely lead to different positions of the neuropilin MAM domain in the semaphorin/plexin/neuropilin complex, suggesting that the MAM domain acts as a part of the spacer and its exact position is not important. This role for the MAM domain is consistent with the observations that it does not interact with either semaphorin or plexin[7,36,44,46].

**Interface between PlexinA4 and Sema3A.** The binding mode between the sema domains of PlexinA4 and Sema3A is highly similar to that seen in the previously reported structures of the semaphorin/plexin complexes, both those dependent and independent of neuropilin (Supplementary Fig. 5)[6,7,9,10,33]. The A106K mutation in Sema3A was designed on the basis that the equivalent position in Sema6A is Lys110, which contributes to the high affinity, neuropilin-independent binding with PlexinA2 through both hydrophobic and electrostatic effects[45]. Our structure confirms that the artificial Lys106 residue in Sema3A indeed plays an analogous role in enhancing the interaction with PlexinA4 but does not alter the binding mode (Supplemental Fig. 5)[6,10].

**Interface between PlexinA4 and Nrp1.** The only binding interface between PlexinA4 and Nrp1 in the cryo-EM structure is mediated by the Sema domain of PlexinA4 and the a1 domain of Nrp1 (Fig. 3). The overall mode of this interface is similar to that seen in the previously reported crystal structure of the Sema3A–PlexinA2–Nrp1 complex[7]. However, a superimposition of the two structures based on the plexin Sema domain shows that the a1 domain of Nrp1 in our cryo-EM structure is further away from plexin by ~4 Å compared with the previous crystal structure (Fig. 3a). There are several clashes between the PlexinA2-Sema domain and the Nrp1-a1 domain in the crystal structure, which could be resolved by a small shift of the a1 domain away from PlexinA2. It is also possible that the binding interfaces of the Nrp1-a1 domain with PlexinA2 and PlexinA4 are slightly different.

The a1 domain of Nrp1 belongs to the $Ca^{2+}$-binding CUB domain family, characterized by a 2-layered β-sandwich fold and a $Ca^{2+}$ binding site formed by two inter-strand loops at one end of the domain (Fig. 3a)[27]. Several positively charged residues from the edge of the PlexinA4-Sema domain, including Lys343, Arg344, and Lys345, make electrostatic interactions with the negatively charged outer surface (including Glu78, Glu126, and Glu128) around the $Ca^{2+}$-binding site in the Nrp1-a1 domain (Fig. 3b). This charge-complementary binding interface is a typical feature of $Ca^{2+}$-binding CUB domain-mediated interactions[27]. The amine group in Lys343 in PlexinA4 is located above the $Ca^{2+}$-binding site and makes contacts with the negatively charged residues at the binding site (Fig. 3b). Meanwhile, the hydrophobic part of the Lys343 sidechain is partially buried in the groove formed by Tyr84, Tyr127, and Glu128 in Nrp1. Lys343 is conserved among class A plexins, suggesting that they all use this residue to interact with the a1 domain of neuropilin (Fig. 3c). In contrast, plexins of other classes, which interact with semaphorin independent of neuropilin, do not have a lysine residue at this position (Fig. 3c).

The interface between the PlexinA4-Sema and Nrp1-a1 domains is rather small, suggesting a weak binding similar to that between PlexinA2 and Nrp1, which has a reported value of dissociation constant of 66 μM[7]. The interaction between full-length plexin and neuropilin is likely stronger on the cell surface where they may reach higher local concentrations and are confined to 2-dimentional space. In addition, it is possible that they may form additional interactions with each other through their transmembrane and intracellular regions.

**Interface between Sema3A and Nrp1-a1.** The Sema domain of Sema3A in our Cryo-EM form a dimer virtually identical to all other dimeric semaphorins. The map for the PSI and Ig-like domains of Sema3A is relatively weak, but allowed a partial model of these domains to be built. The model shows that the two Ig-like domains from each of the two Sema3A subunits tilt toward each other and may contribute to the homodimerization by making contacts through the surface-exposed face of the A–B–E–D β-sheet (Fig. 2a). The structures of Sema4D and Sema7A show similar contributions of the Ig-like domain to the dimerization, but the Ig-like domain in the previous crystal structures of class 3 and class 6 semaphorins is not resolved[6,9,10]. The C-terminal tail following the Ig-like domain, which is not resolved in the Cryo-EM structure, contains the cysteine (Cys723) that forms the interchain disulfide that further stabilizes the Sema3A dimer (Fig. 2a).

The interface between the Nrp1-a1 domain and the Sema domain of Sema3A in the cryo-EM structure is similar to the corresponding interface in the crystal structure of the Sema3A/PlexinA2/Nrp1 complex (Fig. 4). In this case, however, the a1 domain packs more closely with the Sema domain in the Cryo-EM structure. As mentioned above, a small shift of the a1 domain away from plexin and towards semaphorin in the crystal structure would lead to a closer match with the cryo-EM structure. The Sema domain of Sema3A uses the middle segment (residues 375–386) of the long loop between strands 3 and 4 in the fifth blade of the β-propeller to pack against one of the two β-sheets in Nrp1-a1 (Fig. 4b). The interface involves numerous hydrophobic packing interactions, mediated by residues, such as Tyr375 and Phe386 from Sema3A and Tyr44, Pro45 and Pro73 from Nrp1. The interface also involves sidechains of polar residues, including Arg372 from Sema3A, Asn72, His74, and Arg137 from Nrp1.

**Interface between Sema3A and Nrp1-a2.** Previous structural analyses have shown that the a1 domain and the triangle-shaped a2b1b2 module in neuropilin are conformationally independent from each other, owing to the flexible linker between them[7,28]. Consistent with this notion, the Cryo-EM structure here shows yet another relative orientation between the a1 and a2b1b2 domains of Nrp1. The linker between the a1 and a2 domains (residues 141–146) adopts a relatively extended conformation and is oriented roughly orthogonal to the connecting β-strands in the a1 and a2 domains respectively (Fig. 4a). The position of the a2 domain is fixed by its interaction with the Sema domain of Sema3A. The a2 domain is also a $Ca^{2+}$-binding CUB domain. Interestingly, the outer surface of the $Ca^{2+}$-binding site in the a2 domain interacts with Sema3A in a manner that resembles the interface between the Nrp1-a1 and PlexinA4-Sema domains (Fig. 4c). In this case, Lys497 in the Sema domain of Sema3A, corresponding to Lys343 in PlexinA4, is partially buried in the surface groove above the $Ca^{2+}$-binding site and makes electrostatic interactions with Glu195 and Asp250 in Nrp1-a2. Tyr208 in

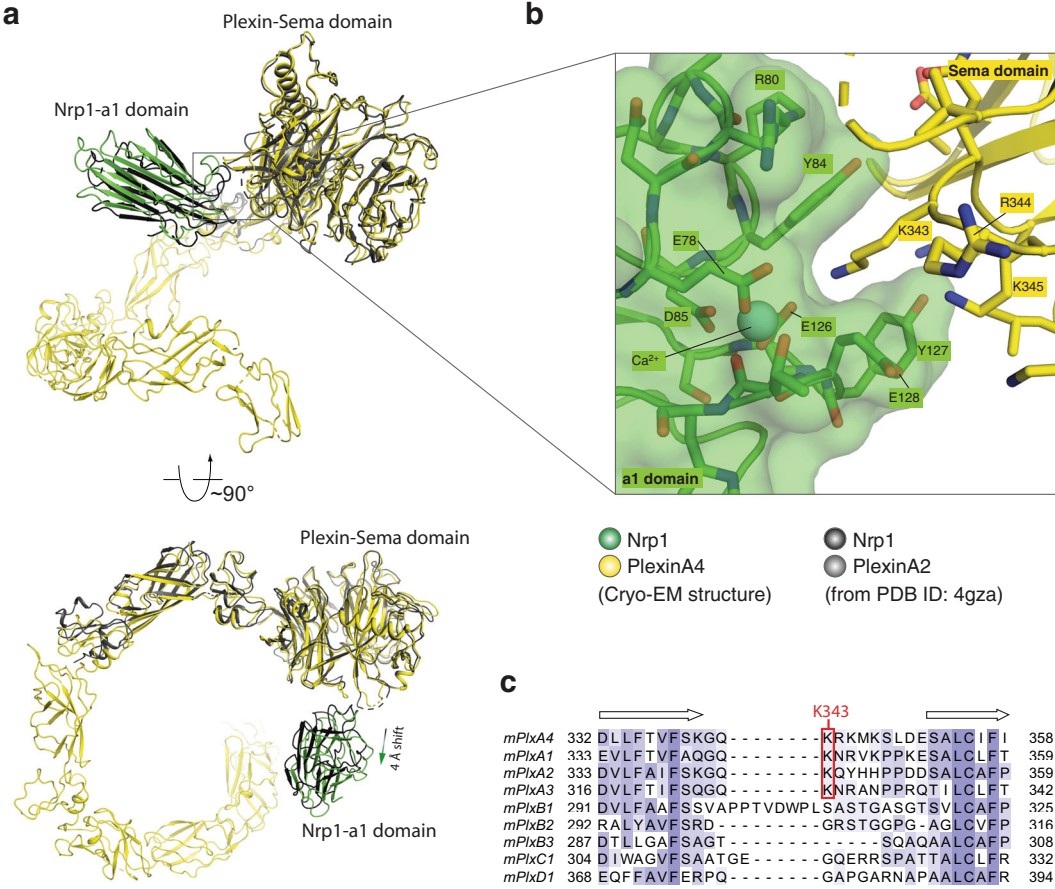

**Fig. 3 Interface between PlexinA4 and Nrp1-a1. a** Overview of the PlexinA4-Sema/Nrp1-a1 interface. For clarity, only one Nrp1-a1 domain and one PlexinA4 molecule from the 2:2:2 complex are shown. One complex of PlexinA2/Nrp1-a1 from PDB ID 4gza is superimposed for comparison. **b** Detailed view of the PlexinA4-Sema/Nrp1-a1 interface. **c** Sequence alignment of the loop region which in mouse PlexinA4 contains Lys343 that is important for interacting with Nrp1-a1.

Nrp1-a2 participates in the interaction by packing with the hydrophobic portion of the Lys497 sidechain. Lys497 is conserved in semaphorins of class 3 but not of other classes (Fig. 4d), suggesting that it is a specificity determinant in the interaction with neuropilin, analogous to Lys343 in class A plexins. There are some additional contacts between a long loop (residues 196–208) near the $Ca^{2+}$-binding site in the a2 domain and the PSI domain of Sema3A, although the weak density around this region prevents more detailed description.

**Interaction between Nrp1-b1 and the C-terminal tail of Sema3A.** The b1 and b2 domains in the a2b1b2 module of Nrp1 are located away from Sema3A. The pocket in the b1 domain that binds the C-terminal arginine residue of semaphorin is ~80 Å away from the Ig-like domain of Sema3A (Fig. 2a). There is a long and presumably flexible linker between the interchain disulfide (Cys723) to the C-terminal arginine residue (R770) (Fig. 2c). Assuming the disulfide bond is located on top of the Ig-like domain at the two-fold symmetry axis of Sema3A, the two C-terminal tails would diverge to opposite directions to reach for the b1 domains of the two Nrp1 molecules. The ~50-residues linker is sufficient to span the distance to allow the C-terminal arginine to engage the binding pocket in the Nrp1-b1 domain. There are several alternative furin-cleavage sites in the tail of class 3 semaphorins, which would generate shorter tails (Fig. 2c). For

example, Sema3A could be cleaved at Arg735 or Arg761[37]. In addition, due to the degeneracy of furin-cleavage motifs and the high abundance of lysine and arginine residues in the Sema3A tail, there are several more potential cleavage sites, including Arg731, Arg734, Arg737, and Arg758 (Fig. 2c). The tail of Sema3A cleaved at Arg761 is sufficiently long and expected to be able to interact with the Nrp1-b1 domain in the same manner as Arg770. However, the tail ending around Arg735 only contains 10–15 residues, which cannot cover the 80 Å distance. There are about 40 residues in the linker between the Ig-like domain (ending at around residue H680) and Cys723 (Fig. 2c). This linker could help one of the C-termini to reach the b1 domain binding pocket on one of the Nrp1 molecules. However, in doing so, the second C-terminus of Sema3A would be dragged further away from the second b1 domain in the 2:2:2 complex, leading to an asymmetric binding mode that is likely less stable. Interestingly, while the furin-cleavage site corresponding to Arg761 is largely conserved in class 3 semaphorins, some of them (Sema3C, Sema3D, and Sema3F) lack the site equivalent to Arg770 (Fig. 2c)[37]. In addition, the length of the C-terminal tail region varies among class 3 semaphorins. The different furin-cleavage patterns of Class 3 semaphorins and alternative usage of the multiple cleavage sites have been shown to play a role in regulating their neuronal repulsive activity during embryonic development[37]. The structural analyses presented here provide a mechanistic explanation for such regulation.

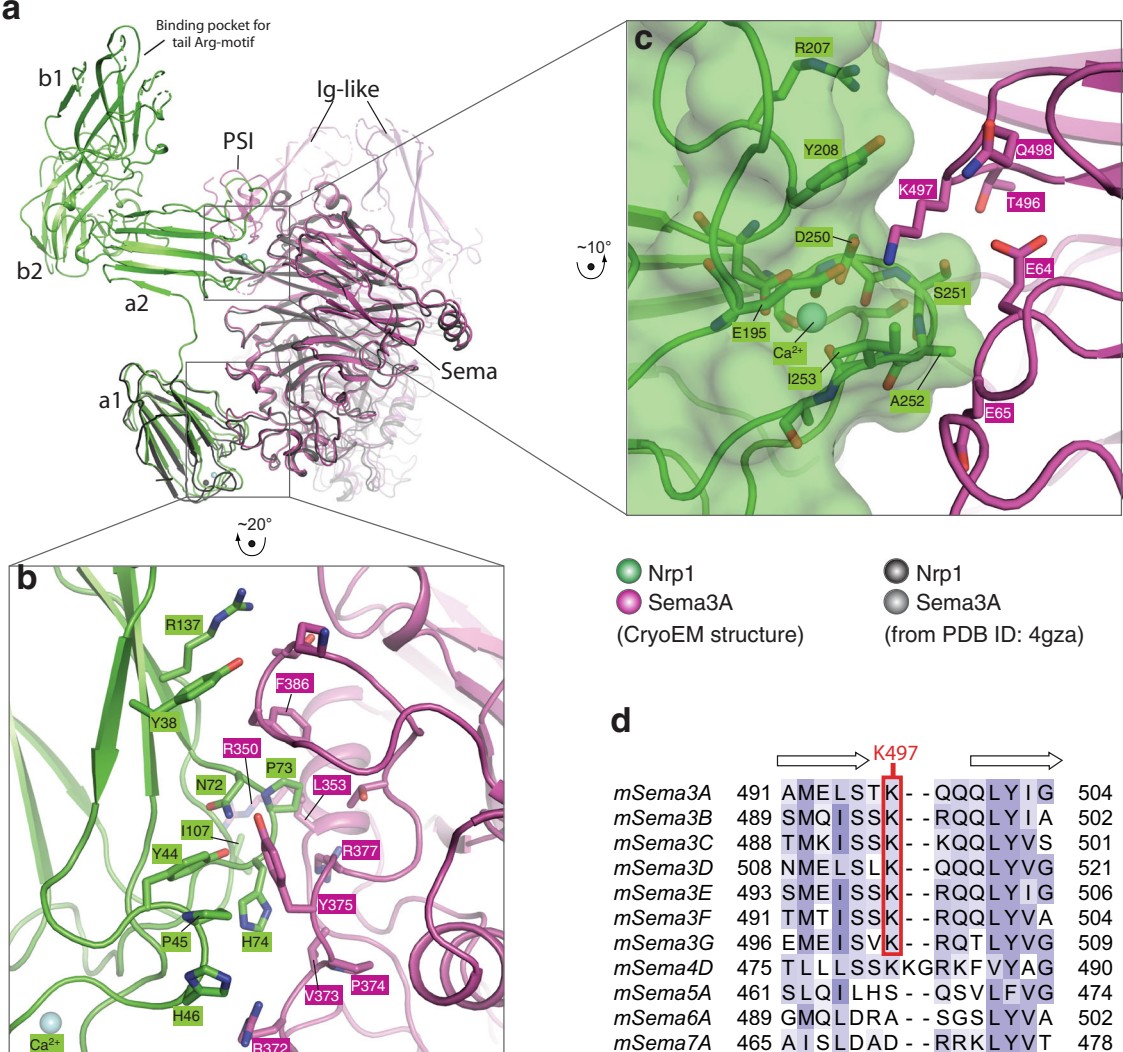

**Fig. 4 Interfaces between Sema3A and Nrp1. a** Overview of the Sema3A/ Nrp1 interfaces. For clarity, only one Nrp1 molecule and the Sema3A dimer from the 2:2:2 complex are shown. One complex of Sema3A/Nrp1-a1 from PDB ID 4gza is superimposed for comparison. **b** Detailed view of the interface between the Sema3A-Sema domain and the Nrp1-a1 domain. **c** Detailed view of the interface between the Sema3A-Sema domain and the Nrp1-a2 domain. **d** Sequence alignment of the loop region which in mouse Sema3A contains Lys497 that is important for interacting with Nrp1-a2.

**Structure-based mutational analyses of the Sema3A/PlexinA4/ Nrp1 complex**. To validate the Cryo-EM structure, we designed mutations based on the structure and examined their effects on both the complex formation and the cellular function of the complex. The complex formation was accessed by an in vitro pull-down assay with purified proteins where FLAG-tagged PlexinA4 extracellular region was used to pull-down Sema3A and Nrp1-a1a2b1b2 (Supplementary Fig. 7). The cellular function was tested by using the well-established COS7 cell collapse assay, in which Sema3A treatment of COS7 cells expressing full-length PlexinA4 and Nrp1 leads to cell morphological collapse that represents semaphorin-induced neuronal growth cone collapse and repulsive guidance[24]. Normal cell surface expression of PlexinA4 and Nrp1 wild-type and mutants was confirmed by immunofluorescence microscopy (Supplementary Fig. 8). We did not test the interface between plexin and semaphorin, because this interface in the Cryo-EM structure is consistent with those in previous studies, which have been extensively examined through mutational analyses[6,9,10]. The Sema3A proteins used in these experiments do not contain the A106K mutation.

To test the interface between PlexinA4 and the Nrp1-a1 domain, we introduced a charge-reversal mutation to Lys343 in PlexinA4, which makes key interactions with the $Ca^{2+}$-binding site in Nrp1-a1 as mentioned above (Fig. 3b). As shown in Supplementary Fig. 7b, PlexinA4-K343E pulled down less Sema3A and Nrp1 compared with PlexinA4-WT. Conversely, a charge-reversal mutation of Glu128 in Nrp1-a1, which interacts with PlexinA4-Lys343, showed a similar detrimental effect as PlexinA4-K343E in the pull-down assay (Supplementary Fig. 7b). Mutation of Arg80 in Nrp1-a1, located at the periphery of the PlexinA4/Nrp1-a1 interface, did not cause obvious decrease in the complex formation (Supplementary Fig. 7c). For the collapse assay, we used Sema3A (21–770) at 5 nM to treat COS7 cells stably expressing full-length PlexinA4 and Nrp1. The results showed that such treatment for 30 min led to ~93% collapse of cells expressing PlexinA4-WT and Nrp1-WT (Fig. 5). In contrast, upon the same Sema3A treatment, only ~2.5% cell expressing PlexinA4-K343E and Nrp1-WT underwent collapse. Likewise, the collapse percentage of cells expressing PlexinA4-WT and Nrp1-E128R was ~8%. These results are qualitatively consistent with the results from the pull-down assay, although the effects of the

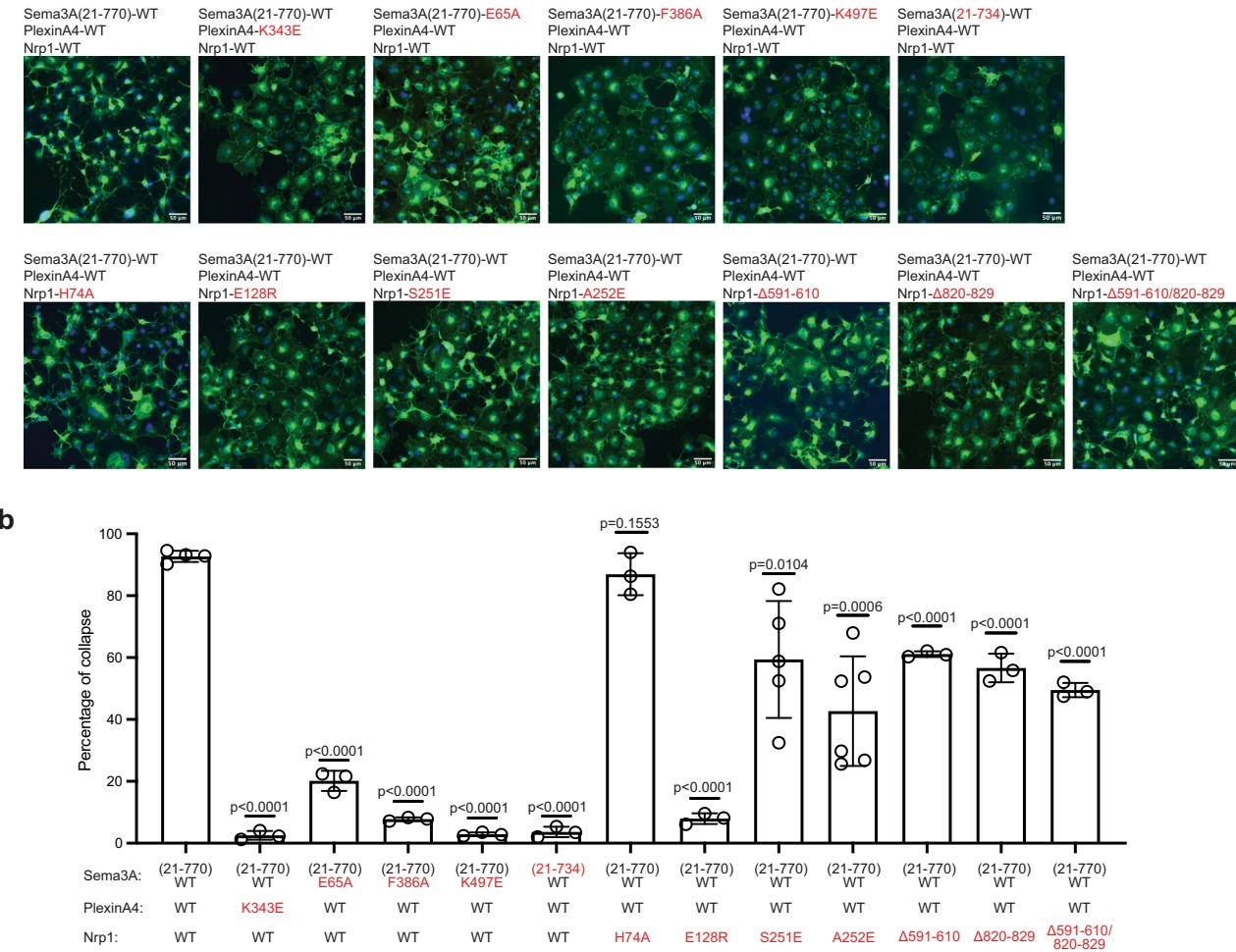

**Fig. 5 COS7 cell collapse assays for mutants of Sema3A, PlexinA4, and Nrp1. a** Images of COS7 cell collapse. Cells stably expressing various combinations of PlexinA4 and Nrp1 constructs were treated with different Sema3A constructs at 5 nM for 30 min. Cells are visualized by immuno-staining for myc-tagged PlexinA4 (green). Nuclei are stained with DAPI (blue). One representative image from at least three biological repeats is shown for each group. **b** Quantification of cell collapse. The individual data points, mean, and s.e.m of the percentage of cell collapse from at least three biological repeats are shown ($n = 4$ for Sema3A-WT/PlexinA4-WT/Nrp1-WT, $n = 5$ for Sema3A-WT/PlexinA4-WT/Nrp1-S251E, $n = 6$ for Sema3A-WT/PlexinA4-WT/Nrp1-A252E, $n = 3$ for other groups). Statistical significance $p$-values between wild-type and each mutant group were determined by two-tailed Student's $t$-test and shown on the top of the bars. Scale bar, 50 μm.

mutations are more pronounced in the collapse assay, suggesting that the latter is more sensitive. Due to the intricate multiple interfaces made by each molecule in the 2:2:2 Sema3A/PlexinA4/Nrp1 complex, individual mutations are expected to reduce but not completely abolish the formation of the complex. Considering this, the vastly different protein concentrations used in the two assays, 5 nM Sema3A and cell surface expressed PlexinA4 and Nrp1 in the cell collapse assay versus μM concentrations of all three proteins in the pull-down assay, likely underlie their different levels of sensitivity to the mutations.

We introduced similar mutations to test the Sema3A-Sema/Nrp1-a2 interface (Fig. 4c). A charge-reversal mutation of Sema3A-Lys497 at the core of the interface reduced the complex formation in the pull-down assay (Supplementary Fig. 7b). Consistently, Seme3A-K497E only induced ~3% collapse of COS7 cells expressing PlexinA4-WT and Nrp1-WT (Fig. 5). On the Nrp1-a2 side of this interface, Ser251 and Ala252 were mutated. The A252E mutant showed clear a detrimental effect in the pull-down assay, whereas S251E did not seem to have an effect (Supplementary Fig. 7c). COS7 cells expressing PlexinA4-WT

and either Nrp1-S251E or Nrp1-A252E showed significantly reduced collapse, ~59% and ~43%, respectively, which again suggest that the collapse assay is more sensitive (Fig. 5).

We also tested in the interface between the Nrp1-a1 domain and the Sema domain of Sema3A (Fig. 4b). An alanine mutation of Sema3A-Phe386, located at the center of this interface, led to reduced formation of the Sema3A/PlexinA4/Nrp1 complex in the pull-down assay (Supplementary Fig. 7b). The H74A mutation of Nrp1 in the Sema3A/Nrp-a1 interface, however, did not cause obvious reduction in the binding. Consistent with these results, while the F386A mutation of Sema3A led to a dramatic reduction in cell collapse to ~8%, the H74A mutation of Nrp1 had no significant effect on cell collapse (Fig. 5).

Our analyses above suggested that the long linker between the disulfide-forming Cys723 and the C-terminal arginine residue in Sema3A acts as a required spacer for the optimal binding to Nrp1 (Fig. 2). To test this hypothesis, we compared the complex formation by two Sema3A constructs, ending at Arg734 and Arg770, respectively. The results from the pull-down experiments showed that Sema3A (21–770) pulled down more PlexinA4 and

Nrp1 than Sema3A(21–734) (Supplementary Fig. 7b). Consistently, in the COS7 cell collapse assays, while Sema3A (21–770) induced ~93% collapse of cells expressing PlexinA4 and Nrp1, Sema3A (21–734) at the same concentration only induced ~4% collapse (Fig. 5). These results are consistent with a previous study showing that, among the various forms of Sema3A, the one ending at Arg770 possessed the highest repulsive guidance activity for cultured sympathetic ganglia from chick embryos[37].

As described above, the b2-MAM linker, MAM and MAM-TM linker regions in neuropilin do not directly mediate interactions in the extracellular region complex, but serve as necessary spacers for the formation of the complex of the full-length proteins on the cell surface. We therefore used the COS7 cell collapse assay to examine the effects of deletions in these regions of Nrp1. We designed a mutant with 20-residue deletion in the b2-MAM linker (Δ591–610), and a mutant with 10-residue deletion in the MAM-TM linker (Δ820–829). A third mutant was generated by combining these two deletions. The results show that these link deletions reduced the percentage of cell collapse to ~50%, significantly lower than that of the wild-type (Fig. 5). Among all the three deletion mutants, the remaining ~70–90 residues in the two linkers in an extended conformation can span the 100 Å distance required for the complex formation on the surface. However, the entropic cost for the mutants to do so is likely increased, which may underlie their reduced but not abolished ability to mediate cell collapse.

Together, the results from the pull-down and cell collapse assays validate our cryo-EM structure and identify many residues that are key to the formation and signaling of the Sema3A/PlexinA4/Nrp1 complex.

## Discussion

While interactions of semaphorin with the plexin receptor and neuropilin co-receptor have been extensively investigated before, the Cryo-EM structure here provide a near complete view of the 2:2:2 extracellular region complex of these three large multi-domain proteins. The overall architecture of the 2:2:2 complex, dictated by multiple relatively weak interfaces contributed by each component, arranges the two plexin molecules in a way that can promote their activation and signaling (Fig. 6). The order in which the three proteins assemble into the 2:2:2 complex could vary as shown in Fig. 6. The heterodimerization of both semaphorin and plexin could substantially diversify the composition of the 2:2:2 complexes, which may gain addition structural features and generate different signaling outputs[8,31,32]. The ring-shape of the 10-domain extracellular region of class A plexins is essential for this activation mechanism, as its curvature allows the two copies of the membrane proximal IPT6 domain to converge and thereby induce the formation of the active dimer of the intracellular region[13,34,47]. The ring-shape has also been shown to mediate the formation of the ligand-independent dimer of class A plexins that prevents spontaneous activation in the absence of the ligand by keeping the intracellular region in the monomeric state (Fig. 6)[47].

As mentioned above, the ring-shape of PlexinA4 appears quite rigid in both the apo-state and the 2:2:2 complex. However, three crystal structures of the full-length extracellular region of PlexinA1 display substantial variations in the ring-shape[33,47]. Docking one of these PlexinA1 structures (PDB ID: 5l59) to the cryo-EM structure based on superimposition of the Sema domain of PlexinA1 and PlexinA4 shows that this conformation of PlexinA1 can form a 2:2:2 active complex similar to that of PlexinA4 (Supplementary Fig. 6b). However, the same modelling of the other two PlexinA1 crystal structures (PDB IDs: 5l56 and 5l5c) results in severe clashes between the two IPT6 domains in the complex, suggesting that these

conformations are not compatible with the complex formation or activation of PlexinA1 (Supplementary Fig. 6b). These observations suggest that the conformational flexibility in the extracellular region of different plexin family members may play a role in regulating their activation. Along this line, the relatively short linker (10–15 residues) between the IPT6 and transmembrane region of class A plexin may allow the transmembrane region to sense the difference in the distance between the two copies of the IPT6 domains in the 2:2:2 semaphorin/plexin/neuropilin complexes, thereby finely tuning the formation of the intracellular active dimer of plexin. This type of regulation in other receptors such as RET, the EGF receptor and c-Kit, has been shown to lead to differences in strength or duration of signaling, and in some cases biased signaling where different downstream pathways are activated to ultimately drive qualitatively distinct biological outcomes[48–51]. Similar regulation of signaling in plexin may allow closely related plexin family members to carry out different functions, despite their overlapping ligand-binding specificities.

The Cryo-EM structure presented here clarifies the roles for the five domains and the interdomain linkers of neuropilin in the formation of the semaphorin/plexin/neuropilin complex. Interestingly, in both neuropilin and semaphorin, long flexible linkers are required for serving as spacers for the formation of the 2:2:2 complex. Such long linkers in proteins are often neglected in structural and functional studies because they are unstructured and nonconserved in sequence. Our analyses suggest that serving as spacers might be a common function of long linkers, especially in large multi-protein assemblies. Related to this point, the MAM domain in neuropilin is not directly involved in binding of semaphorin or plexin, but truncation of this domain abolished the ability of neuropilin in mediating semaphorin-induced neuronal growth collapse[24,36,44,46]. Early co-immunoprecipitation experiments suggested that the MAM domain regulate signaling by mediating dimerization or oligomerization[36,44,46]. However, more recent structural and biophysical analyses have provided strong evidence for lack of dimerization of the MAM domain[28,30]. Consistent with these results, our structure shows that the two MAM domains are placed on opposite sides of the 2:2:2 semaphorin–plexin–neuropilin complex, unlikely to interact with each other. However, the MAM domain in neuropilin is a part of the linker-MAM-linker spacer that is required for the proper formation of the 2:2:2 semaphorin/plexin/neuropilin complex on the cell surface, which at least in part accounts for the functional importance of this domain in neuropilin. A remaining question is whether the linker-MAM-linker region in neuropilin connects to the transmembrane region through the center or outside of the ring of the plexin extracellular region (Fig. 2a). This question also pertains to how the transmembrane regions of plexin and neuropilin are organized and whether they form specific interactions in the 2:2:2 semaphorin/plexin/neuropilin complex. A structure of the complex of full-length semaphorin, plexin, and neuropilin is required for addressing these interesting mechanistic questions.

## Methods

**Protein expression and purification.** For recombinant expression of the extracellular region of PlexinA4, the segment of mouse *PlexinA4* cDNA encoding residues 33–1226 (excluding the N-terminal signal peptide) was subcloned into a modified pETZ-BM vector as described previously[33], which includes the N-terminal signal peptide from alkaline phosphatase and a C-terminal His$_8$-tag (Supplementary Table 1). N-terminal FLAG-tagged PlexinA4 were generated by inserting the FLAG-encoding sequence immediately following the sequence peptide region through PCR reactions. The constructs for mouse Nrp1-a1a2b1b2 (residues 22–588) and Nrp1-a1a2b1b2MAM (residues 22–855) used the same pETZ-BM vector. For expression of mouse Sema3A with a C-terminal arginine residue, representing furin-cleaved mature isoforms, segments encoding residues 21–734 or 21–770 were cloned into another pETZ-BM vector that contains both the signal peptide from alkaline phosphatase and

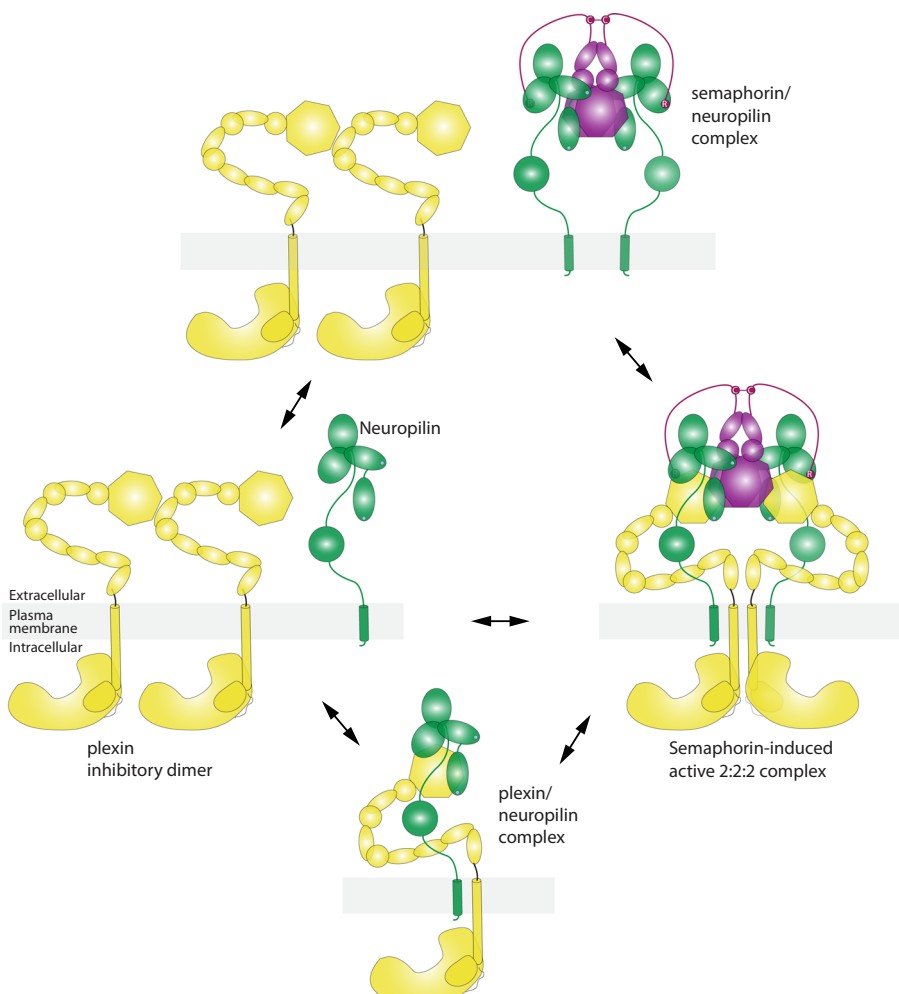

**Fig. 6 Cartoon model of the regulatory mechanisms of class A plexins.** Prior to ligand binding, class A plexins form the inhibitory dimer that prevents the formation of the intracellular active dimer (left panel). It is possible that neuropilin uses its a1 domain to bind plexin, disrupting the inhibitory dimer and priming plexin for activation (lower panel). Neuropilin could bind the semaphorin dimer on its own, considering the strong interaction between the two (upper panel). Ultimately, the semaphorin dimer induces the formation of the 2:2:2 complex, which in turn induces the intracellular active dimer of plexin for signaling (right panel). Whether and how the transmembrane region of plexin and neuropilin interact in the 2:2:2 complex is unclear.

a His6-tag at the N-terminus. Internal furin-cleavage sites in Sema3A, including [551]RRTRR[555], [731]RKQRR[735], and [758]RNRR[761] were mutated to [551]ARTRA[555], [731]AAQAA[735], and [758]ANRA[761], respectively, to prevent cleavage and production of truncated protein. All mutations were generated using PCR-based mutagenesis.

For expression of the individual PlexinA4, Sema3A, and Nrp1 proteins, the respective plasmids were transfected using polyethylenimine (PEI) into Expi293F cells cultured in suspension in Expi293 Expression medium (Thermo-Fisher, Cat#A1435101), with 1000 µg DNA and 4 ml PEI at 1 mg/ml for 1 L cells. For co-expression of the three proteins, the three plasmids at 1:1:1 weight ratio were transfected. Twelve hours post transfection, sodium butyrate with a final concentration of 5 mM were added to Expi293F cells to enhance protein expression. After 7 days expression, medium was collected and exchanged to a buffer containing 25 mM Tris (pH 8.0), 500 mM NaCl, and 25 mM imidazole. The expressed proteins were first purified by Ni-NTA affinity beads using standard protocols. Proteins were further purified by a Superose 6 10/300 gel filtration column (GE healthcare) with a buffer containing 20 mM HEPES (pH 7.5) and 150 mM NaCl and 1 mM $CaCl_2$. Fractions containing the target protein or protein complexes were pooled and concentrated. Purified proteins were directly used for preparing cryo-EM grids or stored at −80 °C for future use. Purified Sema3A (21–770)-A106K, PlexinA4, and Nrp1-a1a2b1b2 could form the complex that remained intact in gel filtration chromatography. Complexes formed with shorter versions of Semaphorin were less stable and tended to dissociate in gel filtration chromatography. Sema3A (21–770)-A106K, PlexinA4, and Nrp1-a1a2b1b2 were co-expressed for convenience as well as taking advantage of the mutual stabilization of the proteins when the complex was formed as soon as they were secreted into the culture medium. During co-expression, Nrp1-a1a2b1b2 expressed

at much higher level than Sema3 (21–770)-A106K and PlexinA4, lead to a large peak of Nrp1 in addition to the complex peak in the gel filtration run (Supplemental Fig. 1).

**Pull-down assay**. The wild-type or the K343E mutant of FLAG-tagged PlexinA4 (20 µg) was incubated with 50 µl anti-FLAG antibody conjugated resin (Sigma, Cat#2220) in a buffer containing 20 mM HEPES (pH 7.5),150 mM NaCl and 1 mM $CaCl_2$ at 4 °C for 30 min. Various pairs of Nrp1 (wild-type or mutants) and Sema3A (wild-type or mutants), 30 µg each, were added and incubated at 4 °C for an additional hour. In parallel, anti-FLAG resin without FLAG-tagged PlexinA4 bound were incubated with wild-type Sema3A or Nrp1 as controls to ensure that they do not interact with the resin nonspecifically. Resin was pelleted by centrifugation, washed with the buffer containing 20 mM HEPES (pH 7.5), 150 mM NaCl, and 1 mM $CaCl_2$ three times. The samples were boiled and resolved by SDS-PAGE. Protein was stained with Coomassie blue R250.

**COS7 cell collapse assay**. The cDNA of full-length mouse *PlexinA4* with a C-terminal c-myc tag were subcloned into a Lentiviral vector pTY with a puromycin-resistance marker (Clontech, Mountain View, CA). The cDNA of full-length *Nrp1* with a C-terminal FLAG-tag were inserted into a Lentiviral vector pTY with a blasticidin S-resistance marker. The mutants of *PlexinA4* and *Nrp1* were generated in these constructs by PCR-based methods. The plasmids were transfected into HEK293T cells (ATCC, Cat#3216) to generate Lentiviral vectors, which were then used to infect COS7 cells (ATCC, catalogue #CRL-1651). COS7 cells were cultured in DMEM medium (Invitrogen) supplemented with 1% (V/V) penicillin/streptomycinb and 10% (V/V) FBS with 5% $CO_2$ at 37 °C. Puromycin (1 ug/ml) and

blasticidin S (5 ug/ml) were used together to select for COS7 cells co-expressing PlexinA4 and Nrp1. Cell surface expression of these proteins was confirmed with immuno-staining and fluorescence microscopy. Myc-tagged PlexinA4 was detected by using a rabbit anti-Myc antibody (Cell Signaling technology, Cat#2278 S; 250× dilution) and an Alexa-488-conjugated anti-Rabbit IgG secondary antibody (Life Technologies, Cat#A111034; 1000× dilution). FLAG-tagged Nrp1 was detected with a mouse anti-FLAG antibody (Bimake, Cat#A5712; 250× dilution) and an anti-Mouse IgG secondary antibody conjugated with the Cy3 fluorophore (Invitrogen, Cat#A10521; 1000× dilution). Nuclei were stained with DAPI (4,6-diamidion-2-phenylindole). A DeltaVision Core microscope was used to image cells.

COS7 co-expressing various combinations of the wild-type or mutants of PlexinA4 and Nrp1 were seeded in six-well plates with $1 \times 10^6$ cells per well and cultured for 12 h. To induce collapse, cells were incubated with Sema3A (21–770) wild-type or mutants at a concentration of 5 nM for 30 min at 37 °C. Cells were then washed with PBS buffer and fixed in 4% paraformaldehyde. Cells were stained for PlexinA4 with an anti-myc antibody (Cell Signaling technology, Cat#2276 S; 5000× dilution) and Alexa-488-coupled anti-mouse IgG secondary antibody (Thermo-Fisher, Cat#A11029; 1000× dilution). Nuclei were stained with DAPI. Images were taken at ×200 magnification with a ZOE fluorescent cell imager (Bio-Rad). Collapsed cells displayed reduced area and star-like morphology, in contrast to the characteristic large and round shape of normal uncollapsed COS7 cells. Counting of collapsed cells was carried out in blind. For each group, at least three biological repeats were done. The number of cells counted in each experiment for different groups was in the range of 379–3020. Plots of percentage of collapse and calculation of p-values were done in Graphpad Prism 9.

**Cryo-EM data collection**. The Sema3A/PlexinA4/Nrp1 complex at 2 mg/ml was applied to a glow-discharged Quantifoil R1.2/1.3 300-mesh gold holey carbon grid (Quantifoil, Micro Tools GmbH, Germany). The grid was blotted under 100% humidity at 4 °C and plunged into liquid ethane with a Mark IV Vitrobot (FEI). The serial EM software was used for the data collection[52]. A 300 kV Titan Krios microscope (FEI) with a K3 Summit direct electron detector (Gatan) for Cryo-EM data collection[52]. The slit width of the GIF-Quantum energy filter was set to 20 eV. Images were recorded in the super-resolution counting mode, dose-fractioned into 30 frames with the dose rate of 1.6 e⁻/Å/frame. The magnification was set to equivalent of the pixel size of 1.08 Å.

**Cryo-EM Image processing and 3D reconstruction**. The image processing and 3D reconstruction processes are outlined in Supplementary Fig. 2. Movie frames were binned by the factor 2, motion corrected and dose weighted by using Motioncorr2 1.1[53]. GCTF 1.06 was used for CTF correction[54]. The rest of the image processing was carried out using RELION 3.1[55]. The initial round of particle picking was carried out using the Laplacian of Gaussian blob detection implemented in Relion. Particles were extracted and subjected to several rounds of 2D classification. A subset of particles from good 2D classes were selected for another around of 2D classification. Nine good 2D classes averages from this step were then used for reference-based particle autopicking in Relion. Particles were then extracted with a binning factor of 4 and subjected to three rounds of 2D classification. Particles in good 2D classes were selected and subjected to 3D classification. The initial model was generated by combining the crystal structures of the partial Sema3A/PlexinA2/Nrp1 complex (PDB ID: 4gza) and the apo-full-length extracellular region of PlexinA4 (PDB ID: 5l5k), and converting it to a density map low-pass filtered to 30 Å resolution. A total of 137,836 particles from two good 3D classes were re-extracted to the original pixel size and subjected to 3D refinement the C2 symmetry. The resulted 3D reconstruction reached 4.0 Å overall resolution, but the density for the peripheral parts of the complex was much weaker than the center portion. A round of 3D classification without image alignment of the refined particles led to one class showing stronger density for the peripheral parts. Particles in this class (26741 particles) were selected and subjected to further 3D refinement, CTF refinement, leading to a final 3D reconstruction of resolution 3.7 Å (Table 1 and Supplemental Fig. 3). Resolution was estimated by applying a soft mask around the protein density. The Fourier Shell Correlation (FSC) 0.143 criterion was used. Local resolution was calculated in Relion (Supplemental Fig. 3).

**Model building, refinement, and validation**. Model building was initiated by rigid-body docking of individual domains from the crystal structures of the mouse PlexinA4 extracellular region (PDB ID: 5l5k), Sema3A (PDB ID: 4gz8), and the a1a2b1b2 domains Nrp1 (PDB ID: 2qqk). Manual building was carried out using the program Coot 0.89[56]. The final model does not include sidechain for many residues, the C-terminal tail of Sem3A and several interdomain loops of the three proteins due to lack of clear density. The model was refined by using the real-space refinement module in the Phenix package (V1.8)[57]. Restraints on secondary structure, backbone Ramachandran angels, residue sidechain rotamers were used during the refinement to improve the geometry of the model. MolProbity 4.5 as a part of the Phenix validation tools was used for model validation (Table 1)[58]. Figures were generated in PyMOL 2.3 (Schrödinger, LLC, 2015, the PyMOL Molecular Graphics System) or Chimera 1.3[59]. Sequence alignments are rendered with JalView 2.11.1.3[60].

**Reporting summary**. Further information on research design is available in the Nature Research Reporting Summary linked to this article.

## Data availability
The atomic coordinates and cryo-EM map have been deposited into the RCSB (entry ID: 7M0R) and EMD database (entry ID: EMD-23613), respectively. All the relevant data are available from the authors. Source data are provided with this paper.

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

## Acknowledgements

Cryo-EM data were collected at the University of Texas Southwestern Medical Center (UTSW) Cryo-Electron Microscopy Facility, funded in part by the Cancer Prevention and Research Institute of Texas (CPRIT) Core Facility Support Award RP170644. We thank Dr Daniel Stoddard and Jose Martinez Diaz for facility access. We thank Hongtao Yu and the Structural Biology Lab at UTSW for equipment usage. This work is supported in part by grants from the National Institutes Health (R35GM130289 to X.Z. and R01GM136976 to X.-C.B.), the Welch foundation (I-1702 to X.Z. and I-1944 to X.-C.B.), and CPRIT (RP160082 to X.-C.B.). X.-C.B. and X.Z. are Virginia Murchison Linthicum Scholars in Medical Research at UTSW.

## Author contributions

X.Z., G.S., and X.-C.B. conceived the project. D.L., G.S., and X.H. developed the protein expression and complex formation procedures. D.L. and G.S. prepared the cryo-EM samples. X.-C.B. and X.Z. collected the cryo-EM data and solved the structure. D.L. did the binding and cell-based assays. X.Z. wrote the paper with inputs from other authors.

## Competing interests

The authors declare no competing interests.
