## [Peer Review File · Nature Communications]

Reviewers' Comments:

Reviewer #1:

Remarks to the Author:

Semaphorin-Plexin signaling system controls many cellular events that involve cell migration and plays fundamental roles in neuronal, immune, cancer, and bone cells. Semaphorin 3A is a prototypical semaphorin ligand that had been most extensively studied among ~20 semaphorins. Unlike other semaphorin classes, class 3 semaphorins are not membrane-anchored but are secreted. Despite its high biological and medical importance and long history of intensive researches by experts, mechanistic understanding of sema3A's molecular action has been incomplete. Part of the reason for this difficulty stemmed from the fact that it undergoes multiple processing by furin-like proteases and is labile after isolation, has very low affinity toward its signaling receptor Plexin-A, and requires non-signaling co-receptor neuropilin for its biological activity on cells. A low resolution crystal structure exists for a tripartite complex made of sema3A, plexinA2, and neuropilin, all as minimally truncated fragment, but the incomplete nature of this structural information precluded us from elucidating how these three proteins will cooperate to execute rapid and sensitive signal transduction on the cell surface.

In this paper, Kuo et al present the 3.7Å-resolution cryo-EM structure of the sema3A-PlexA4-Nrp full signaling complex that can explain many of the unresolved puzzles. Through many structural studies of semaphorin-plexin complexes, including those from the current authors, it is already accepted that engagement by homodimeric semaphorin ligand will draw two plexin molecules into symmetric dimer, leading to the activation of the cytoplasmic GAP domain to activate downstream signaling. The same plexin active dimer is also visualized in the current structure; however, in the current intertwined tripartite 2:2:2 complex it is stabilized by interfaces provided by four-domain Nrp, including the ones that had not been appreciated before. Importantly, the critical importance of the sema3A Arg770 that had been postulated to function as the ligand for Nrp b1 domain is now better explained in the structural context, although not directly visualized in the structure.

The paper is very well written, clear and concise, and the experiments (including sample prep, EM analysis, and functional assays using cells) were performed at high standard. Although the resolution of the EM map is of modest quality, this reviewer found no overfitting or overinterpretations in the final model, and the method section contains sufficient details about the EM analysis procedure. Overall, I think this fine paper should be published without unnecessary delays. I only have minor comments/suggestions that might be useful to improve the paper.

1. As this is a paper describing experimental determination of a cryo-EM structure, I feel it rather odd that the authors only show structural models (both ribbons and surfaces) as the main figures. At least I would like to see the experimental EM map (e.g. Suppl Fig.4a) as part of the main figure.
2. The authors discuss the potential of differently processed sema3A molecules to engage Nrp b1 domain using the C-terminal Arg (page 15). Specifically, they mention that the version ending with Arg735 may engage only one of the Nrp, leading to a less stable asymmetric complex. I wonder whether they have some experimental data for this, since they produced sema3A protein encoding residues 21-734 (method section, page 23). What did you find when this shorter version of sema3A was co-expressed with PlexA4 and Nrp?
3. The cartoon model in the Fig. 7 seems to suggest that all three components (sema3A dimer, Nrp, and Plexin) will encounter and form 2:2:2 complex at the same time, but if we consider the affinity difference, Nrp must capture sema3A first, forming 2:1 or 2:2 complex prior to the recruitment of low-affinity plexin. The authors may want to consider modifying this figure to incorporate this point.
4. In the page 15, line 12, "binding pocket in the Nrp1-b2 domain" should be Nrp1-b1 domain?

(end of comments)

reviewed by Junichi Takagi

Reviewer #2:

Remarks to the Author:

The authors of this manuscript have determined the structure of what they believe is the sema3A

signal transducing complex which they presume to contain sema3A neuropilin-1 and plexin-A4. To do that they combined the soluble extracellular domains of neuropilin-1 and plexin-A4 in the presence of sema3A. In contrast with a prior study (Janssen et al) which used a furin truncated form of sema3A which leads to a much lower bioactivity of sema3A (Adams et al 1997) and which solved a structure that contained plexin-A2 rather than plexin-A4, in this study the authors used indeed a longer sema3A construct that is indeed fully active in cytoskeleton collapse assays.

This approach yielded some novel insights with regard to the structure of the complex that is formed between these constituents in solution. The authors went on to show the functional importance of their findings by the introduction of point mutations in key locations of the three proteins and show nicely that these mutations perturb the formation of the complexes. Furthermore, signal transduction is perturbed when COS7 cells over expressing np1 and plexin-A4 are challenged with sema3A containing such mutations or when mutated receptors are expressed in the cells and challenged with wild type sema3A.

I am not a structural biologist and therefore I do not feel that I am qualified to pass judgment on the technical parts that concern complex formation and data interpretation. I do have some concerns that pertain to the overall design of the study. I do realize that for the structural studies very high concentrations of sema3A, neuropilin-1 extracellular domain and plexin-A4 extracellular domain were used and that it may not be possible to conduct such studies under different conditions. However, these conditions do not recapitulate the in-vivo situation in which there exist much lower concentrations of these receptors. Plexin-A4 in particular is usually expressed in most cells at very low concentrations. Thus the design of the present study ignores prior studies which have shown that in primary endothelial cells as well as in other cell types signal transduction initiated by sema3A which results in cytoskeletal collapse requires not only plexin-A4 but also plexin-A1 (Kigel et. al. 2011, Sabag et. al. 2014). These studies indicate that a signaling complex containing only plexin-A4 may not enable signal transduction at the plexin-A4 concentrations usually found in living cells. The authors test their constructs by over expressing plexin-A4 and neuropilin-1 in COS7 cells. It is possible that the COS7 cells used here may express enough endogenous plexin-A1 which may be sufficient to enable signal transduction in conjunction with plexin-A4. The authors should in my opinion verify that there is no plexin-A1 or plexin-A2 is expressed by the cells they used, preferably by knocking-out the plexin-A1 gene in the COS7 cells using CRISPR/Cas9 in order to make sure that the signal transduction they see is indeed mediated by a complex containing only sema3A, neuropilin-1 and plexin-A4. In addition, these studies show, for example, that plexin-A2 can replace plexin-A1 in the signaling complex but that much higher concentrations of plexin-A2 are required in that case as compared to the concentration of plexin-A1. It is thus possible that a sufficiently high non-physiological concentration of plexin-A4 may abrogate the requirement for plexin-A1 and thus the authors should make sure that the expression levels of neuropilin-1 and plexin-A4 in the COS7 cells are similar to the concentrations encountered naturally in cells such as in endothelial cells.

Another major point of concern is the structure of the sema3A used in the study. The authors state that "We also introduced to the semaphorin constructs the mutation corresponding to A106K in Sema3A, which was rationally designed to enhance the binding to plexin (Gioelli et al., 2018)". This mutation was introduced in the Gioelli paper because it increases the affinity of sema3A to plexin-A4 and in that paper even enables neuropilin independent activation of plexin-A4 mediated signaling by sema3A. In my opinion it is incorrect to introduce such a mutation into work that attempts to study the structure of the natural functional sema3A receptor as it may lead to erroneous conclusions. The authors should verify that their results are also valid when using unmodified sema3A. While the modification of the furin cleavage sites can be justified since as far as we know they do not change fundamentally the interaction of full length sema3A with its receptors, this is not the case for the A106K mutation. It was not clear to me from the text if the sema3A constructs used in the collapse assays in Figure 6 contained the mutation. If they did, than in my opinion the experiments should be repeated using sema3A that does not contain the A106K mutation.

Reviewer #3:

Remarks to the Author:

Lu et. al. reported a structure of Sema3A/PlexinA4/Neuropilin complex, which is a more complete view of this tripartite complex as it comprises of almost entire extracellular domains of the three components. Overall, the structure was determined using standard processing pipeline, properly validated, and further supported by previously determined crystal structures of subcomplexes. The biochemical and cell based assays confirmed experimentally observed interfaces and key molecular interactions. My minor suggestions are listed below:

1. In Figure 5, changes in band intensity are hard to tell visually. It may help to quantify using image processing software. Also, this figure could be moved to supplemental data as binding data are not as sensitive as cell-based assays.
2. I am not doubting expertise of the authors, but it might help to use TOPAZ particle-picking software to get cleaner particles for subsequent data processing. To be clear, I am not the developers of TOPAZ and do not suggest that reprocessing is required for publication of this manuscript.

Reviewer #1 (Remarks to the Author):

Semaphorin-Plexin signaling system controls many cellular events that involve cell migration and plays fundamental roles in neuronal, immune, cancer, and bone cells. Semaphorin 3A is a prototypical semaphorin ligand that had been most extensively studied among ~20 semaphorins. Unlike other semaphorin classes, class 3 semaphorins are not membrane-anchored but are secreted. Despite its high biological and medical importance and long history of intensive researches by experts, mechanistic understanding of sema3A's molecular action has been incomplete. Part of the reason for this difficulty stemmed from the fact that it undergoes multiple processing by furin-like proteases and is labile after isolation, has very low affinity toward its signaling receptor Plexin-A, and requires non-signaling co-receptor neuropilin for its biological activity on cells. A low resolution crystal structure exists for a tripartite complex made of sema3A, plexinA2, and neuropilin, all as minimally truncated fragment, but the incomplete nature of this structural information precluded us from elucidating how these three proteins will cooperate to execute rapid and sensitive signal transduction on the cell surface.

In this paper, Kuo et al present the 3.7Å-resolution cryo-EM structure of the sema3A-PlexA4-Nrp full signaling complex that can explain many of the unresolved puzzles. Through many structural studies of semaphorin -plexin complexes, including those from the current authors, it is already accepted that engagement by homodimeric semaphorin ligand will draw two plexin molecules into symmetric dimer, leading to the activation of the cytoplasmic GAP domain to activate downstream signaling. The same plexin active dimer is also visualized in the current structure; however, in the current intertwined tripartite 2:2:2 complex it is stabilized by interfaces provided by four-domain Nrp, including the ones that had not been appreciated before. Importantly, the critical importance of the sema3A Arg770 that had been postulated to function as the ligand for Nrp b1 domain is now better explained in the structural context, although not directly visualized in the structure.

The paper is very well written, clear and concise, and the experiments (including sample prep, EM analysis, and functional assays using cells) were performed at high standard. Although the resolution of the EM map is of modest quality, this reviewer found no overfitting or overinterpretations in the final model, and the method section contains sufficient details about the EM analysis procedure. Overall, I think this fine paper should be published without unnecessary delays. I only have minor comments/suggestions that might be useful to improve the paper.

Response: We thank this reviewer for these positive comments.

1. As this is a paper describing experimental determination of a cryo-EM structure, I feel it rather odd that the authors only show structural models (both ribbons and surfaces) as the main figures. At least I would like to see the experimental EM map (e.g. Suppl Fig.4a) as part of the main figure.

Response: This is a great suggestion. We have moved Supplemental Fig. 4a to the new Figure 1.

2. The authors discuss the potential of differently processed sema3A molecules to engage Nrp b1 domain using the C-terminal Arg (page 15). Specifically, they mention that the version ending with Arg735 may engage only one of the Nrp, leading to a less stable asymmetric complex. I wonder whether they have some experimental data for this, since they produced sema3A protein encoding residues 21-734 (method section, page 23). What did you find when this shorter version of sema3A was co-expressed with PlexA4 and Nrp?

Response: We indeed tried these different versions of Sema3A for the formation of the complex, through both co-expression and mixing of purified components. Complexes formed with shorter versions of Sema3A, including 21-734, were less stable and often did not co-elute in gel filtration chromatography. We therefore did not pursue structure analyses of these proteins. We have added this description in the method section.

3. The cartoon model in the Fig. 7 seems to suggest that all three components (sema3A dimer, Nrp, and Plexin) will encounter and form 2:2:2 complex at the same time, but if we consider the affinity difference, Nrp must capture sema3A first, forming 2:1 or 2:2 complex prior to the recruitment of low-affinity plexin. The authors may want to consider modifying this figure to incorporate this point.

Response: This is great suggestion. We have modified the figure accordingly and added two sentences in the discussion section to describe this point. In the modified figure, we also depict the possibility that plexin and neuropilin may form the receptor/co-receptor complex on the cell surface prior to semaphorin binding.

4. In the page 15, line 12, "binding pocket in the Nrp1-b2 domain" should be Nrp1-b1 domain?

Response: Thanks for spotting this error. It has been corrected now.

(end of comments)

reviewed by Junichi Takagi

Reviewer #2 (Remarks to the Author):

The authors of this manuscript have determined the structure of what they believe is the sema3A signal transducing complex which they presume to contain sema3A neuropilin-1 and plexin-A4. To do that they combined the soluble extracellular domains of neuropilin-1 and plexin-A4 in the presence of sema3A. In contrast with a prior study (Janssen et al) which used a furin truncated form of sema3A which leads to a much lower bioactivity of sema3A (Adams et al 1997) and which solved a structure that contained plexin-A2 rather than plexin-A4, in this study the authors used indeed a longer sema3A construct that is indeed fully active in cytoskeleton collapse assays.

This approach yielded some novel insights with regard to the structure of the complex that is formed between these constituents in solution. The authors went on to show the functional importance of their findings by the introduction of point mutations in key locations of the three proteins and show nicely that these mutations perturb the formation of the complexes. Furthermore, signal transduction is perturbed when COS7 cells over expressing np1 and plexin-A4 are challenged with sema3A containing such mutations or when mutated receptors are expressed in the cells and challenged with wild type sema3A.

Response: We thank this reviewer for these positive comments.

I am not a structural biologist and therefore I do not feel that I am qualified to pass judgment on the technical parts that concern complex formation and data interpretation. I do have some concerns that

pertain to the overall design of the study. I do realize that for the structural studies very high concentrations of sema3A, neuropilin-1 extracellular domain and plexin-A4 extracellular domain were used and that it may not be possible to conduct such studies under different conditions. However, these conditions do not recapitulate the in-vivo situation in which there exist much lower concentrations of these receptors. Plexin-A4 in particular is usually expressed in most cells at very low concentrations. Thus the design of the present study ignores prior studies which have shown that in primary endothelial cells as well as in other cell types signal transduction initiated by sema3A which results in cytoskeletal collapse requires not only plexin-A4 but also plexin-A1 (Kigel et. al. 2011, Sabag et. al. 2014). These studies indicate that a signaling complex containing only plexin-A4 may not enable signal transduction at the plexin-A4 concentrations usually found in living cells. The authors test their constructs by over expressing plexin-A4 and neuropilin-1 in COS7 cells. It is possible that the COS7 cells used here may express enough endogenous plexin-A1 which may be sufficient to enable signal transduction in conjunction with plexin-A4. The authors should in my opinion verify that there is no plexin-A1 or plexin-A2 is expressed by the cells they used, preferably by knocking-out the plexin-A1 gene in the COS7 cells using CRISPR/Cas9 in order to make sure that the signal transduction they see is indeed mediated by a complex containing only sema3A, neuropilin-1 and plexin-A4. In addition, these studies show, for example, that plexin-A2 can replace plexin-A1 in the signaling complex but that much higher concentrations of plexin-A2 are required in that case as compared to the concentration of plexin-A1. It is thus possible that a sufficiently high non-physiological concentration of plexin-A4 may abrogate the requirement for plexin-A1 and thus the authors should make sure that the expression levels of neuropilin-1 and plexin-A4 in the COS7 cells are similar to the concentrations encountered naturally in cells such as in endothelial cells.

Response: The concentrations of the proteins used in the structural analyses were indeed much higher than those on the cell surface. As recognized by this reviewer, this is inevitable with the current state-of-the-art in structural biology. The structural analyses presented here therefore cannot capture some of the subtle yet important aspects of plexin signaling. This reviewer correctly pointed out that we failed to discuss one important mechanism in plexin signaling, that is, some class 3 semaphorins require the simultaneous presence of two different class A plexins for signaling, especially under physiological expression levels (much lower than those in overexpression experiments). These observations suggest that heterodimers of plexin have unique structural and functional properties that are not present in homodimers. We have added a few sentences in both the introduction and the discussion to discuss this point and included relevant references. That said, we believe that the results from the cell-based assays are sufficient validation for the binding interfaces shown by the cryo-EM structure, as loss of cell collapse caused by the mutations supports the importance of the mutated residues in the complex formation.

This reviewer suggested two interesting scenarios that could alter semaphorin signaling activity and specificity in the cell-based assay. It is possible that either one or both scenarios were involved in our experiments. Nevertheless, they do not affect our interpretation of the results, which were designed to show the importance of the residues in the binding interfaces in the structure. We agree that the CRISPR knockout experiments suggested by this reviewer could reveal some interesting mechanistic insights. However, we respectfully submit that they are out of the scope of this paper. We do hope to follow up on this point in future studies. In particular, it would be very interesting to solve structure of those heterodimeric complexes, and design mutations to understand why they show additional signaling activities than homodimeric complexes.

Another major point of concern is the structure of the sema3A used in the study. The authors state that "We also introduced to the semaphorin constructs the mutation corresponding to A106K in Sema3A, which was rationally designed to enhance the binding to plexin (Gioelli et al., 2018)". This mutation was introduced in the Gioelli paper because it increases the affinity of sema3A to plexin-A4

and in that paper even enables neuropilin independent activation of plexin-A4 mediated signaling by sema3A. In my opinion it is incorrect to introduce such a mutation into work that attempts to study the structure of the natural functional sema3A receptor as it may lead to erroneous conclusions. The authors should verify that their results are also valid when using unmodified sema3A. While the modification of the furin cleavage sites can be justified since as far as we know they do not change fundamentally the interaction of full length sema3A with its receptors, this is not the case for the A106K mutation. It was not clear to me from the text if the sema3A constructs used in the collapse assays in Figure 6 contained the mutation. If they did, than in my opinion the experiments should be repeated using sema3A that does not contain the A106K mutation.

Response: We agree that this is valid concern. However, as shown in Supplementary Figure 5, the binding interface formed by the Sema3A-A106K mutant with plexin is nearly identical to those in other plexin/semaphorin complexes, strongly suggesting that this mutation does not affect the binding mode, but merely increases the binding affinity for plexin. We have added a few words in the text to clarify this point. We thank this reviewer for pointing out the importance of using the wild type Sema3A, rather than the A106K mutant, in the pull-down assays and the collapse assays. We indeed used Sema3A that does not contain the A106K mutation in these experiments, but the text of the manuscript did not describe it explicitly. We have added a sentence highlight this important point at the end of the first paragraph of this section.

Reviewer #3 (Remarks to the Author):

Lu et. al. reported a structure of Sema3A/PlexinA4/Neuropilin complex, which is a more complete view of this tripartite complex as it comprises of almost entire extracellular domains of the three components. Overall, the structure was determined using standard processing pipeline, properly validated, and further supported by previously determined crystal structures of subcomplexes. The biochemical and cell based assays confirmed experimentally observed interfaces and key molecular interactions. My minor suggestions are listed below:

Response: We thank this reviewer for these positive comments.

1. In Figure 5, changes in band intensity are hard to tell visually. It may help to quantify using image processing software. Also, this figure could be moved to supplemental data as binding data are not as sensitive as cell-based assays.

Response: The changes in band intensities are indeed difficult to see for some mutants, because each protein makes multiple interactions with the others in the complex and single mutations does not completely disrupt its formation. We did repeat these experiments many times, and found that the results are qualitatively reproducible. The following image shows a side-by-side comparison of the original Figure 5a (now Supplemental figure 7b) and an independent repeat. However, pull-down experiments are not quantitative in nature, and complicated normalization schemes must be used in order to average different repeats. We therefore feel that quantification of gels does not add much value to the figure. Following the suggestion of this reviewer, we have moved this figure to the supplemental data (Supplemental figure 7).

**Gel shown in original Fig 5a
(now Fig S7b)**

A different repeat of the same set of mutants

2. I am not doubting expertise of the authors, but it might help to use TOPAZ particle-picking software to get cleaner particles for subsequent data processing. To be clear, I am not the developers of TOPAZ and do not suggest that reprocessing is required for publication of this manuscript.

Response: We thank this reviewer for this suggestion. With Cryo-EM technology developing so quickly, it is always possible that new methods could provide better structures. As suggested, we tried particle-picking using TOPAZ, and Cryolo as well. In this case, TOPAZ worked better (we had some previous cases where Cryolo worked better). However, the reconstruction from the particle set generated by TOPAZ is still a bit worse than what we report in the manuscript. Based on our experience with these proteins in the past few years, we believe that dissociation and damaging of the complex during EM-grid preparation were the major reasons why the number of good particles were small. We hope to improve in this respect in future.